# Exploring the Cost-Effectiveness of Newborn Screening for Metachromatic Leukodystrophy (MLD) in the UK

**DOI:** 10.3390/ijns10030045

**Published:** 2024-06-26

**Authors:** Karen Bean, Simon A. Jones, Anupam Chakrapani, Suresh Vijay, Teresa Wu, Heather Church, Charlotte Chanson, Andrew Olaye, Beckley Miller, Ivar Jensen, Francis Pang

**Affiliations:** 1Orchard Therapeutics, London W6 8PW, UK; charlotte.chanson@orchard-tx.com (C.C.); andrew.olaye@orchard-tx.com (A.O.); francis.pang@orchard-tx.com (F.P.); 2Division of Cell Matrix Biology & Regenerative Medicine, School of Biological Sciences, Faculty of Biology, Medicine and Health, Manchester University NHS Foundation Trust, Manchester M13 9WL, UK; simon.jones@mft.nhs.uk (S.A.J.);; 3Great Ormond Street Hospital, London WC1N 3JH, UK; 4Birmingham Women’s and Children NHS Foundation Trust, Birmingham B4 6NH, UK; 5Precision AQ, Boston, MA 02108, USA; beckley.miller@precisionaq.com (B.M.); ivar.jensen@precisionaq.com (I.J.)

**Keywords:** cost-effectiveness, cost utility analysis, metachromatic leukodystrophy, MLD, newborn screening, NHS, decision-tree model

## Abstract

Metachromatic leukodystrophy (MLD) is a fatal inherited lysosomal storage disease that can be detected through newborn bloodspot screening. The feasibility of the screening assay and the clinical rationale for screening for MLD have been previously demonstrated, so the aim of this study is to determine whether the addition of screening for MLD to the routine newborn screening program in the UK is a cost-effective use of National Health Service (NHS) resources. A health economic analysis from the perspective of the NHS and Personal Social Services was developed based on a decision-tree framework for each MLD subtype using long-term outcomes derived from a previously presented partitioned survival and Markov economic model. Modelling inputs for parameters related to epidemiology, test characteristics, screening and treatment costs were based on data from three major UK specialist MLD hospitals, structured expert opinion and published literature. Lifetime costs and quality-adjusted life years (QALYs) were discounted at 1.5% to account for time preference. Uncertainty associated with the parameter inputs was explored using sensitivity analyses. This health economic analysis demonstrates that newborn screening for MLD is a cost-effective use of NHS resources using a willingness-to-pay threshold appropriate to the severity of the disease; and supports the inclusion of MLD into the routine newborn screening programme in the UK.

## 1. Introduction

Metachromatic leukodystrophy (MLD) is an ultra-rare and fatal inherited lysosomal storage disease caused by a deficiency of arylsulfatase-A (ARSA), due to mutations in the *ARSA* gene. Reduced ARSA activity results in the accumulation of sulfatides in the CNS and peripheral nervous system, which leads to progressive demyelination, rapid motor and cognitive decline and premature death, particularly in early-onset MLD (late infantile [LI] and early juvenile [EJ]). The incidence of MLD in the UK has been estimated to be 1 case in 40,000–160,000 live births and the clinical spectrum of MLD is broad [1,2]. However, four clinical forms are commonly described on the basis of age at first symptoms’ onset, the most severe being late infantile (LI, onset of symptoms < 30 months), which typically presents between 15 and 24 months and is the most common variant, accounting for 50–60% of all MLD cases in the UK; the other three include early juvenile MLD (EJ, onset of symptoms between 30 months and <7 years), late juvenile (LJ, onset of symptoms between 7 and <17 years) and adult-onset MLD (AO, ≥17 years) [3]. The late infantile form is rapidly progressive and characterised primarily by delays in gross and fine motor function, delays in and loss of speech, muscular hypotonia, ataxia and spasticity, which subsequently progresses to complete loss of motor and cognitive functions. The early juvenile presentation and progression of MLD is more variable, but once patients have lost the ability to walk independently, the deterioration of gross motor and cognitive function is as rapid as the infantile form [4,5,6]. Earlier age at onset or presence of motor symptoms at onset is associated with a more severe and rapid disease course. For example, the median time from onset of symptoms to the inability to walk independently, characterised by being beyond the Gross Motor Classification System for MLD (GMFC-MLD) stage 1 has been reported to be 2.7 months for LI patients and approximately 7 months for EJ patients [6]. Furthermore, the prognosis for patients without treatment is very poor—the median time from onset of symptoms to beyond GMFC-MLD 4, which can be characterised by the complete loss of any locomotion or being able to sit without support, has been reported to be only 1.12 years for LI patients and 2.7 for EJ MLD patients [6]. The fact that the same study reported a median survival for LI patients of 8.42 years, and 68.6% of EJ patients were alive 15 years from symptom onset [6], implies that untreated early-onset MLD patients (defined as late infantile and early juvenile MLD) spend the majority of the remainder of their lives with significant morbidity. This leads to an extremely high caregiving burden for MLD [7] and also has a significant impact on healthcare resource use in the NHS. 

Prior to the regulatory approval of atidarsagene autotemcel (arsa-cel; OTL-200), treatment of early-onset MLD in the UK consisted only of best supportive care (BSC), as no disease-modifying treatments had been approved. BSC aims to manage disease complications and maintain quality of life for as long as possible but does not target the root cause of the progressive motor and cognitive decline. Best supportive care includes and is not limited to: physical therapy to maintain mobility; muscle relaxant medications to reduce spasticity; pain management; management of skeletal deformity; respiratory physiotherapy to manage pulmonary infections; anti-convulsant drugs to control seizures; anti-psychotic medications to control psychiatric symptoms; dietary support; enteral nutrition through a feeding tube; and family and psychological counselling. However, patients still experience rapid disease progression ultimately ending in a severely debilitated state, with the median age of death reported in the literature to be 4.2 and 9.86 years of age for the most common MLD phenotype, the late infantile form [6,8,9]. Haematopoietic stem cell transplantation (HSCT) is used as a treatment option for the late juvenile or adult-onset MLD phenotypes in UK clinical practice; but it is not considered a viable option for early-onset MLD by the UK clinicians with expertise in treating MLD. 

Arsa-cel is an autologous CD34+ haematopoietic stem and progenitor cell (HSPC) gene therapy. HSPCs are collected from mobilised peripheral blood (mPB) and transduced ex vivo with a lentiviral vector (ARSA LVV), which inserts one or more copies of the human *ARSA* complimentary deoxyribonucleic acid (cDNA) into the cell’s genome, so that genetically modified cells become capable of expressing the functional ARSA enzyme. Arsa-cel uses a lentiviral vector which allows the gene to be integrated directly into the genome of the target cell, where it can be replicated whenever the cell divides. As such, the added gene is subsequently passed on to all of the cell’s progeny. Furthermore, the self-renewal capability of HSPCs suggests that once the genetically corrected HSPCs successfully engraft in the brain, there would be a durable supply of the genetically corrected cells and their progenies. Arsa-cel received regulatory approval in the European Union in December 2020 and January 2021 in the UK, for the treatment of MLD characterised by biallelic mutations in the *ARSA* gene leading to a reduction of the ARSA enzymatic activity: in children with late infantile or early juvenile forms without clinical manifestations of the disease (pre-symptomatic—PS-LI and PS-EJ), and in children with the early juvenile form with early clinical manifestations of the disease, who still have the ability to walk independently and before the onset of cognitive decline (early symptomatic—ES-EJ) [10]. Furthermore, in February 2022 arsa-cel was recommended for reimbursement by the National Institute for Health and Care Excellence (NICE) in the UK for its licensed indication based on the provision of a patient access scheme and with the assurance that treatment be administered in a highly specialised service by a specialist multidisciplinary team [11]. 

In stark contrast to the published data for BSC [4,5,6], the published clinical data for arsa-cel show that with up to approximately 11 years of follow-up in 39 patients, arsa-cel provides meaningful clinical benefit to patients with early-onset MLD treated in both the pre-symptomatic and early symptomatic stages of the disease [3,12]. Published data show that treatment with arsa-cel resulted in sustained, clinically relevant benefits in children with early-onset MLD by preserving cognitive function and motor development in most patients, and slowing demyelination and brain atrophy [3,12]. Analysis of two symptomatic EJ patients included in the clinical trial who experienced rapid disease progression after gene therapy showed that levels of engraftment and pharmacodynamic effects post-gene-therapy were within the range observed in patients that responded to treatment, and no differences were observed between the two groups of patients with respect to percentage of LV+ cells, vector copy number (VCN) in the bone marrow (BM), VCN in peripheral blood mononuclear cells (PBMCs) or ARSA activity in peripheral blood and cerebral spinal fluid (CSF). However, the two ES-EJ patients (defined as treatment failures) were treated at symptomatic stages of the disease when disease progression was entering the rapid progressive phase and would not be eligible for treatment according to the eligibility definition of the populations suitable within the EU and UK indication for arsa-cel [3,10]. These data are important because they highlight the narrow window of time for treatment and the importance of identifying and treating early-onset MLD as soon as possible. 

In the absence of newborn screening (NBS) in current clinical practice in the UK, pre-symptomatic MLD patients are only identified if an older sibling is affected, and due to the rapidly deteriorating nature of MLD, the window for treatment (before advanced nervous system damage occurs) can be limited—often only months. Without NBS, UK clinical experts in the management of MLD estimate that only about 15–20% of all early-onset MLD patients’ diagnoses will occur within the treatment window for arsa-cel (these patients will predominantly be siblings of older affected children), and worryingly, given that it is the most common phenotype, it is highly unlikely that an index case of late infantile MLD would ever be diagnosed pre-symptomatically. In fact, all the pre-symptomatic late infantile patients included in the clinical trial of arsa-cel were diagnosed early and enrolled only because of an older affected sibling, who him/herself could not be offered treatment due to advanced disease [3]. Clinical expert opinion was corroborated by recently published real-world evidence (RWE), which examined MLD disease phenotype, presentation, eligibility and affected siblings in the year following NHS approval of arsa-cel using hospital records [13]. The authors found that of 17 UK MLD patients who were referred for treatment, only four patients met eligibility criteria and were treated. Eleven patients failed screening, consisting of 10 symptomatic patients with late infantile disease and 1 with early juvenile disease and cognitive decline. Two further patients with later-onset subtypes did not meet the approval criteria. Three out of the four treated patients were diagnosed by screening after MLD was diagnosed in a symptomatic older sibling, and the other patient was an early-symptomatic early juvenile patient who was diagnosed within the treatment window [13]. The authors noted that the vast majority (79%) of children referred for gene therapy were deemed ineligible due to advanced disease [13].

The importance of early diagnosis and initiation of treatment is now increasingly recognised for a number of rare genetic conditions and inborn errors of metabolism, with the most beneficial response to treatment seen in patients prior to the onset of symptoms [14]. For MLD specifically, this is because patients treated in the early symptomatic stage of the disease will have incurred some irreversible cell damage as a result of the accumulated sulfatides. Whilst the sulfatides are broken down by the newly restored ARSA enzyme, the damage done persists throughout the patients’ lifetime, such that the quality of life enjoyed by these patients is not perceived to be as great as that of those patients who show no manifestations of the disease. Therefore, when MLD patients are treated with arsa-cel pre-symptomatically and in advance of predicted symptom onset, arsa-cel has the potential to address the motor and cognitive aspects of disease progression and allow children to have improved length and quality of life compared to untreated patients and patients treated in the early symptomatic stages of the disease. 

Due to the recent approval of a gene therapy for early-onset MLD, newborn screening using a single punch from the routine dried blood spot (DBS) sample has become a feasible option to maximise the best possible outcomes for all MLD patients, where previously only late juvenile or adult-onset MLD patients had an effective treatment option. Pilot studies using a newly developed two-tier screening algorithm for MLD are underway in multiple countries globally, including the UK. Published data from one of these de-anonymised newborn screening pilots in the US in over 27,000 newborn dried blood spots demonstrated near 100% assay specificity [15]. To minimise the false-positive rate, the authors designed the two-tier screening algorithm such that the primary test was to quantify C16:0-sulfatide in DBS by ultraperformance liquid chromatography–tandem mass spectrometry (UPLC-MS/MS). The screening cut-off for this was established based on the results from 15 MLD newborns to achieve 100% sensitivity. The secondary test was to measure the ARSA activity in DBS from newborns with abnormal C16:0-sulfatide levels. Only newborns that displayed both abnormal C16:0-sulfatide abundance and ARSA activity were considered screen positives. Results showed that of the 27,335 samples screened, 2 high-risk cases were identified. ARSA gene sequencing identified these two high-risk subjects to be an MLD-affected patient and a heterozygote [14]. Recently published data from a prospective study in Germany demonstrated proof of concept for the feasibility of a high-throughput method for newborn screening for MLD using the same sulfatide and ARSA enzyme-based algorithm. As of December 2023, 120,000 babies have been screened, with three cases of MLD identified [16]. Specific to the UK, results from a pre-pilot study showed that the two-tier C16:0-S and ARSA screening strategy was successfully validated using day-5 newborn bloodspots, with high specificity for MLD; using a cut-off of 160 nmol/L for the sulfatides correctly identified all 19 positive MLD day-5 bloodspots [17]. 

If treatment for MLD is administered prior to the onset of symptoms in screen-positive babies, it is imperative that the prediction of MLD subtypes in pre-symptomatic neonates is possible. European consensus-based recommendations on the clinical management of newborn screening in MLD strongly recommend predicting the age of symptom onset based on family history, genotype and ARSA enzyme activity [18]. In neonates without affected relatives, the prediction relies primarily on genotype–phenotype correlations as published in the literature and public databases. These databases indicate whether the mutations are null (0) or residual (R) and are updated regularly. The general rule is that a 0/0 genotype is most likely to be late infantile; and for 0/R and R/R mutations, EJ MLD is suspected if it is based on known mutations which have been shown to cause EJ-MLD, e.g., the mutation of the ARSA variant c.465+1G>A coupled with the second most frequent variant c.1283C>T, p.(Pro428Leu). Homozygosity for the c.1283C>T, p.(Pro428Leu) variant is associated with late-juvenile or adult onset [18].

There are several published studies examining the cost-effectiveness of newborn screening for other inborn errors of metabolism, but not MLD. Bessey et al. examined the cost-effectiveness of including five additional inborn errors of metabolism (glutaric aciduria type 1, homocystinuria, isovaleric acidaemia, long-chain hydroxyacyl CoA dehydrogenase deficiency and maple syrup urine disease) in the UK Newborn screening programme to provide evidence to the UK National Screening Committee [14]. The authors found that screening for all of the conditions was more effective and cost-saving when compared to not screening for each of the conditions. As a consequence, these conditions were added to the national screening programme. In addition, a recent study by Weidlich et al. demonstrated that newborn screening for SMA was a cost-effective use of NHS resources in the UK, where health outcomes for patients with SMA were improved with NBS and less costly when compared to no screening [19]. 

For NBS for MLD to be adopted nationally, there must be evidence that it is cost-effective. Therefore, this study aimed to evaluate whether it is cost-effective to add MLD to the routine UK newborn screening dried blood spot (DBS) programme using a decision analytic framework from a national health service/personal social services perspective.

## 2. Materials and Methods

### 2.1. Model Inputs—Epidemilogy and Chance Node Probabilities

Using a model framework similar to that adopted in Bessey et al. [14] and Weidlich et al. [19], a series of decision-tree models were constructed using Microsoft Excel for each MLD phenotype, and are depicted schematically in Figure 1. The long-term incremental costs and outcomes were then calculated for the screening vs. no-screening arms proportionally to the epidemiological data for the phenotypes (Table 1). Model inputs for the epidemiology of MLD and arsa-cel treatment eligibility were based on published data and the clinical experience of metabolic disease experts from the three major UK specialist MLD hospitals: Manchester University NHS Foundation Trust, Manchester, UK, Great Ormond Street Hospital, London, UK and the Birmingham Women’s and Children’s NHS Foundation Trust, Birmingham, UK (Table 1). Data for these inputs were generated through clinical expert group discussion and consensus gained during a workshop intended to gather the evidence required to calculate the cost-effectiveness of newborn screening for MLD in the UK.

In the no-screening branch of the model, separate arms for patients with a family history of MLD vs. no family history were constructed to reflect the very different probabilities of receiving treatment and the time of treatment in these two groups (Table 2). For example, where there is a family history of MLD, i.e., there is an older affected sibling (index case), subsequent children will pro-actively be tested for MLD and consequently will be identified before the onset of symptoms, increasing the likelihood they will be eligible for treatment and accrue the positive long-term health outcomes. In the screening arm, the probability of a confirmed positive screen, negative screen, false positive and false negative were based on published literature (Table 3) [15]. For a false negative screening result, the probability of being eligible for treatment with arsa-cel was assumed to be the same as the no-family-history probability (Table 3).

### 2.2. Model Inputs—Long-Term Outcomes

The inputs for the lifetime costs and health outcomes (quality-adjusted life years [QALYs]) denoted by the terminal nodes in Figure 1 for early-onset MLD, arsa-cel and best supportive care were extracted from the health economic model used in the NICE highly specialised technology (HST18) appraisal of arsa-cel [11] and clinical trials [3]. A discount rate of 1.5% for both costs and benefits was applied because arsa-cel has the potential to restore patients who would otherwise die or have a very severely impaired life to full or near-full health, which is likely to be sustained over a very long period (normally at least 30 years). Consequently, the cost-effectiveness analyses are sensitive to the discount rate and in these cases, NICE considers a non-reference case discount rate of 1.5% if it is highly likely that, on the basis of the evidence presented, the long-term health benefits are likely to be achieved. 

The de novo cost-effectiveness model used in the HST appraisal was developed based on an eight-state partitioned survival framework and Markov structure to model disease progression in MLD using the seven-stage Gross Motor Function Classification MLD scale (GMFC-MLD) (Figure 2) using Microsoft Excel. The GMFC-MLD is a validated measure of motor dysfunction in MLD, which represents all clinically relevant stages from normal (GMFC-MLD 0) to loss of all gross motor function (GMFC-MLD 6). For the LI MLD population, there were eight possible health states, grounded by the seven-stage GMFC-MLD health states and a death state, because the rate of motor decline occurs in conjunction with cognitive decline. However, for the EJ population, three cognitive substates (normal, moderate, severe) were included for each of the GMFC-MLD health states to reflect the fact that cognitive decline may not occur at the same rate as motor function loss. Monthly cycles were implemented to capture the rapid deteriorating nature of the disease. 

Based on the clinical data for arsa-cel, the modelling of patient response was grouped into three distinct categories: percentage of full responders—patients who did not demonstrate any signs and symptoms of disease across multiple outcome measures (GMFC-MLD; GMFM; cognitive function [DQp]; MRI, etc.) and were equivalent to the GMFC-MLD 0 health state until their last follow-up (at least 3 years of follow-up and up to 12 years); percentage of stable partial responders—patients who initially experienced some disease progression prior to stabilisation of disease, but who then remained within that health state for the duration of the time horizon of the model; and finally the percentage of unstable partial responders who experienced continued disease progression despite treatment, but at a slower rate than was reported for patients in the natural history arm (BSC). Figure 3 visually depicts the time spent in each GMFC-MLD health state for best supportive care and arsa-cel for the pre-symptomatic late infantile, pre-symptomatic early juvenile and early-symptomatic early juvenile MLD cohorts that were used to calculate the long-term costs and QALYs in both the screening and no-screening arms. Transition probabilities for each of the three responder groups by early-onset MLD subtype can be found in the Appendix A.

Utility values to calculate QALYs associated with the long-term benefits and disease progression arms were generated from a vignette study conducted in the UK general population (*n* = 201). In accordance with established methodologies, a proportion of all the possible health states vignettes (24/28) were valued using lead trade-off method (LTO). The health-state vignettes captured all the clinically relevant stages of MLD from normal (GMFC–MLD 0 with normal cognition) to loss of all gross motor function and cognitive ability (GMFC-MLD 6 with severe cognitive impairment) [21]. Age-adjusted UK general population utility values were used for GMFC-MLD 0 with normal cognition. Utility values from the vignette study were then derived for all health states using the rescaled methodology, which used a simple algorithm involving the EQ-5D tariff specific to the country of interest to reflect the societal preferences for that country. The UK general population evaluated the quality of life of patients in the later stages of disease to be worse than death, generating negative utility values [11]. Health states worse than death were permitted to adequately capture the extremely high burden of disease and the value set was considered appropriate for decision-making [11]. The utility values by GMFC-MLD health state used in the Markov model are presented in Table 4. 

In addition, a caregiver disutility was included for patients in GMFC-MLD 3 to 6 to capture the negative effect on health-related quality of life that caring for an MLD patient has on their families. This disutility was calculated from the mean index utility value (0.773) for all respondents (*n* = 21 in total including UK, US and German respondents) completing the EQ-5D in the MLD Caregiver Survey [22] who are caring for MLD children at different stages in their disease, but predominantly in the latter stages, subtracted from UK general population utility at 40 years of age (0.882), which equates to a disutility of −0.108. This disutility is assumed to last up until the death of the patient or up to a maximum of 30 years, as adult MLD patients are less likely to be cared for at home and more likely to have residential care. As the disease progresses, the caregiver burden increases [11]. The caregiver disutility was applied by number of carers required by GMFC-MLD health state and is detailed in Table 5.

Per-patient total QALYs for the no-screening branch were calculated based on the time spent in each health state in the health economic model of arsa-cel vs. best supportive care used in the NICE HST appraisal (see Figure 3). The total per-patient QALYs accrued are different when patients are treated pre-symptomatically vs. early-symptomatically. This is because patients with early symptoms have already sustained a degree of irreversible motor damage prior to treatment that, as it manifests over time, is associated with a perceived lower quality of life—for example, patients treated with early symptoms may continue to have some worsening of disease prior to stabilization that means they will need aids to walk (GMFC-MLD 2) or require a wheelchair (GMFC-MLD 3). In addition, data from the arsa-cel clinical trials showed that a proportion of the pre-symptomatic late infantile patients were treated close to the predicted onset of symptoms and, consequently, during the engraftment window sustained some motor dysfunction prior to stabilizing in either the GMFC-MLD 1 or 2 normal cognitive function health states (see Figure 3), which are associated with a lower utility value than the health state GMFC-MLD 0—considered equivalent to the healthy general population (see Table 4).

For the percentage of incident MLD patients in the decision-tree model whose disease has progressed too far to be eligible for treatment with arsa-cel, the total QALYs from the BSC arms from the de novo cost-effectiveness model for each of the MLD phenotypes were used to estimate long-term health outcomes at the terminal nodes.

In the screening arm of the decision-tree model, all early-onset MLD patients identified through newborn screening are assumed to be treated within the first 6–8 months of life and well before the predicted onset of symptoms. Based on the clinical data for patients who were treated with arsa-cel well before the predicted onset of symptoms, the positive long-term health outcomes for these patients have been assumed to be equivalent to those calculated for the full-responder group in the no-screening arm i.e., treated patients remain in GMFC-MLD 0 for the duration of the time horizon of the model (see Figure 3). 

In the absence of any published literature on the short-term impact on QoL of a false positive newborn screen result, an arbitrary removal of 0.01 QALYs from the normal health total QALYs in the screening arm has been assumed to reflect the very short-term negative impact on quality of life that a false positive screen result has (a matter of weeks). 

Given that arsa-cel is only indicated for early-onset MLD, for the small proportion (10%) of late-onset MLD patients, additional inputs for lifetime costs and QALYs for treatment with allogeneic hematopoietic stem cell transplant (HSCT) or bone marrow transplant were obtained from the HSCT/BMT arm of the NICE HST7 appraisal of Strimvelis, another gene therapy approved for the treatment of ADA-SCID, as a proxy for MLD in the absence of any published cost-effectiveness data for adult-onset MLD [23]. Whilst late-onset patients would not be treated until sub-clinical evidence of disease was present [18], for the purposes of the modelling the long-term costs of HSCT have been applied assuming late-onset patients will receive HSCT as an infant. In the no-screening arm, for untreated patients it has been assumed that patients will experience normal health until onset of symptoms aged 16. However, in the screening arm an arbitrary whole QALY per patient has been removed to reflect the potential disutility associated with knowing from birth that the child will develop MLD later in life. 

Normal health QALYs were calculated from the healthy general population from age 0–81 using age-specific utility scores from the UK EQ-5D tariff and discounted at 1.5%. 

### 2.3. Costs

The annual cost of adding the MLD screening assay to the current newborn screening programme was calculated as follows: the cost of the first-tier test to detect abnormal C16:0-sulfatide levels is based on the marginal cost of expanding the current existing Tandem Mass Spectrometry (TMS) NHS Newborn Bloodspot Screening system to include this test. Published data from 2013 looking at expanding the national programme to include an additional five inborn errors of metabolism estimated this to be £0.50 per baby for the additional laboratory time [14], which would be £0.71 with this cost inflated to 2023 prices. To calculate the marginal cost of adding the laboratory time to screen for MLD only, this cost was divided by five. Only babies that display abnormal C16:0-sulfatide levels in the first-tier test undergo the second-tier test to examine abnormal ARSA enzyme activity levels from the dried blood spot samples. Given there is a requirement to return to the DBS sample, the cost is no longer marginal, and so the second-tier test has been estimated by geneticists in the labs undertaking the UK pilot study to cost around £4.50 per test. Based on the probabilities outlined in Table 3 of obtaining a positive first- and second-tier screening test, the total annual cost of screening 704,328 newborn babies for MLD is £122,625, which equates to £0.17 per baby.

For the long-term costs, healthcare resource use (HCRU) for each of the GMFC-MLD states in the de novo economic model were obtained from an elicitation exercise with five UK clinical experts in MLD, which were used to estimate the frequency, duration and proportion of HCRU for MLD patients, including medical visits, equipment use and social care use. The unit costs for the individual HCRU items were derived from multiple sources, including the National Cost Collection data, the Personal Social Services Research Unit (PSSRU) costs of health and social care and National Schedule of NHS Costs. The monthly medical costs by GMFC-MLD health state are presented in Table 6. 

Treatment and administration costs for arsa-cel for the early-onset MLD patients and allogeneic HSCT for late-onset patients were also included for all treated patients. Administration costs for both arsa-cel and HSCT included leukapheresis (cell harvest), conditioning, administration and hospitalization, as well as follow-up transplant costs. Treatment and administration costs are presented in Table 7. Whilst the list price of arsa-cel is provided in Table 7, the negotiated price that the NHS pays for arsa-cel was used in the economic model. 

The total lifetime costs assumed for the late-onset MLD are also detailed in Table 7 and are calculated from the average cost reported from the matched unrelated donor HSCT arm and the HSCT haploidentical results from the HST7 economic model. Costs for normal health are assumed to be zero in both decision arms.

### 2.4. Willingness-to-Pay (WTP) Thresholds 

The National Screening Committee (NSC) makes the decision on whether a new screening programme should be recommended in the UK. Typically, the NSC relies on the £20,000 to £30,000 willingness-to-pay (WTP) threshold set by NICE as this is a commonly used standard. In 2022, NICE updated its process and methods guide for health technology evaluation in the UK [25]. Whilst NICE has set the maximum willingness-to-pay (WTP) threshold of £20,000–30,000/QALY for standard appraisals, i.e., those appraisals not within the highly specialised technologies programme (HST), it has replaced the end-of-life criteria, which had a willingness-to-pay threshold of £50,000/QALY, with different QALY weights determined by the severity of disease. The severity modifiers use the absolute and proportional QALY shortfall to determine the QALY weights, where the absolute QALY shortfall is determined by the total number of future QALYs lost due to a condition, and the proportional QALY shortfall is the proportion of future QALYs lost due to a condition. A proportional shortfall of <0.85 has a QALY weight of 1 (WTP £30,000/QALY), ≥0.85 <0.95 has a QALY weight of 1.2 (WTP £36,000/QALY) and ≥0.95 has a QALY weight of 1.7 (WTP £50,000/QALY). Due to the rapidly deteriorating nature of MLD and the very young age of the children affected by the disease who die prematurely, the proportional QALY shortfall for MLD patients is >0.95, garnering a willingness-to-pay threshold of £50,000/QALY gained. Therefore, in order to determine the cost-effectiveness of NBS for MLD, a willingness-to-pay threshold of £50,000 per QALY gained has been used. It is important to distinguish between the technology and the screening programme. Arsa-cel is the technology and has been evaluated under the highly specialised technologies appraisal and is eligible for the higher QALY weights relevant to that programme. However, here, the focus is on evaluating the screening programme itself and the allocation of existing resources across the broader healthcare system. Therefore, the standard technology QALY weighting with the severity modifiers is applicable. The NSC has yet to make a screening decision using the severity modifiers, and it will be at the Committee’s discretion as to whether they adopt this approach.

## 3. Results

### 3.1. Patient Numbers

Based on an incidence rate of 1/100,000 live births, approximately 7 babies would be born with MLD in the UK annually based on an average number of 704,328 live births per year. Table 8 tracks the numbers of patients through both decision arms of the model. The results show that in the screening arm, nearly three times the number of babies are diagnosed with MLD early enough to be eligible for treatment vs. the no-screening arm. This is driven by the very low number of patients with no family history of MLD that are still able to be treated by the time they are diagnosed; this is particularly apparent in the late infantile cohort where less than 1% of LI babies born with no family history will be identified pre-symptomatically and thus be eligible for treatment. Since its approval in the UK, two early-onset MLD patients have been treated with arsa-cel per year, which corroborates the estimated patient numbers in the no-screening arm. 

### 3.2. Base Case Analyses

The introduction of NBS for MLD over the lifetime of a newborn MLD cohort identified per year was associated with an incremental gain of 246 discounted QALYs vs. no screening for MLD. Discounted incremental costs for screening vs. no screening were circa £8.16 million over the lifetime of a newborn MLD cohort identified per year. The incremental cost-effectiveness ratio (ICER) for screening vs. no screening for MLD was £33,212/QALY gained, which is a cost-effective use of NHS resources based on the willingness-to-pay threshold appropriate for MLD of £50,000/QALY gained. Table 9 details the total costs and total QALYs generated per year for the individual branches of the decision-tree model for each MLD subtype, the healthy general population and for all 704,328 newborns dependent on the decision to choose screening or no screening. 

If the QALY outputs from Table 9 are examined in more detail, in terms of the general population/normal health branches, there is only a 0.07 difference in the total number of QALYs in favour of the no-screening arm, which is logical given that only a very small proportion of patients in the screened branch are likely to experience a false positive test, and the potential disutility associated with this is very short-term. 

Examination of the QALYs in the no-screening arm for the ES-EJ cohort show that the net QALY gain is negative. This is as a result of the fact that a large number of patients are not diagnosed within the therapeutic window for arsa-cel, and consequently the majority of ES-EJ patients accrue the QALYs associated with BSC, which, given the amount of time untreated EJ patients spend in the latter stages of the disease in health states that are perceived to be worse than death by members of the UK general population, leads to negative QALY gains. In addition, the maximum health outcomes achievable for treated ES-EJ patients are valued at a lower quality of life than those possible for pre-symptomatic patients who either remain asymptomatic or have some mild motor impairment. Consequently, the QALY gains for the small number of treated ES-EJ patients per year (0.12 persons with NFH and 0.29 persons with FH) are not enough to offset the HRQoL for the majority of ES-EJ patients that receive BSC in the no-screening arm, and therefore the net total QALYs for this cohort are negative.

### 3.3. Exploration of Uncertainty

Table 10 reports results from scenario analyses designed to test the impact that varying key parameters has on the base case cost-effectiveness results, for example, the chance node probabilities and the long-term costs and QALYs.

The input with the greatest impact on the results is the discount rate, as changing the discount rate from 1.5% to 3.5% increases the ICER to above the £50,000/QALY willingness-to-pay threshold. This is not surprising given that most of the costs are incurred upfront, but the benefits accrue over a lifetime. One of the parameters with the second greatest impact on the ICER is the probability that patients will experience disease progression despite treatment in the screening arm. Given that patients are expected to be treated well before the predicted onset of symptoms, it has been assumed that the vast majority will remain in the GMFC-MLD 0 health state for the duration of the time horizon of the model based on the clinical trial results. Reducing this probability by increasing the percentage of patients that experience disease progression despite treatment worsens the ICER because the cost of treatment is incurred for 100% of patients but the long-term benefits are diminished to only 80% of the cohort. Increasing the proportion of patients that experience disease progression prior to treatment in the screening arm has a marginally positive impact on the ICER; this is because although total QALYs are reduced in the screening arm, total costs are also reduced as these patients will not incur treatment costs.

The scenario analyses testing the uncertainty in the MLD incidence rate show that if more babies are born with MLD per year than has been assumed in the base case (1/100,000), the cost-effectiveness of screening for MLD improves. For example, if the RWE reported in Horgan et al. [13] is substituted into the model, the ICER for screening vs. no screening improves to £31,231/QALY gained. Conversely, if fewer babies are born with MLD than assumed in the base case, then the ICER worsens. It is worth noting, however, that even using an extremely low incidence rate of 1/160,000 cases of MLD, the ICER is still below the willingness-to-pay threshold of £50,000/QALY gained. The base case analyses assume that 60% of all MLD cases are late infantile based on published data from Wang et al. [8]; sensitivity analyses varying this probability and the other subtypes show that the distribution of MLD subtypes in the UK has minimal impact on the ICER. 

The marginal cost of adding the first-tier sulfatide test to the existing tandem mass spectroscopy (TMS) tests to analyse heel prick bloodspots was derived from Bessey et al. and adjusted to 2023 prices. The second-tier ARSA enzyme activity assay was estimated by geneticists in the laboratories undertaking the UK pilot study to cost approximately £4.50 per test. Two sensitivity analyses were conducted simultaneously to test uncertainty in the cost of screening. Firstly, the total cost of screening (first tier and second tier) was varied by +/−20%, which had a minimal impact on the ICER (+/−£114). Secondly, given that the cost of the ARSA enzyme assay was an estimated cost, a sensitivity analysis varying this cost by +/−80% was conducted (screening cost scenario a). Results from this analysis demonstrated minimal impact on the ICER (see Table 10); this is because so few newborns require the second-tier part of the screen, as only positive C16:0-sulfatide screens would undergo the second-tier ARSA enzyme activity screen. Given that the cost of the first-tier sulfatide test was calculated from the marginal cost of adding five new screening programmes divided by five, which could be an underestimation of the marginal cost for this test, an additional analysis was conducted in which the screening cost for the first-tier sulfatide test was increased to the marginal cost for adding all five conditions from Bessey et al., adjusted to 2023 prices, which increased the ICER by £2280/QALY gained (screening cost scenario b). 

In order to reflect uncertainty in the long-term outcomes in the screening arm, a combination of a 20% reduction in the incremental QALY gains and a 20% increase in the incremental costs for all the MLD patients in this arm has been explored to establish an arbitrary upper bound for the ICER; in addition, for the lower bound, a 20% increase in the incremental QALY gains and a 20% decrease in incremental costs have been applied to explore a potential lower bound for the ICER of screening vs. no screening. No changes were made to the no-screening arm. As expected, reducing the incremental QALY gains in the screening arm by 20% in combination with increasing costs worsens the ICER for screening vs. no screening to £49,818/QALY gained. Conversely, increasing the QALY gains by 20% and reducing the costs improves the ICER for screening vs. no screening to £22,141/QALY gained. Whilst neither of these two sensitivity analyses reflect individual parameter uncertainty or indicate what percentage of sampling within the parameter distributions in a probabilistic sensitivity analysis would fall below the £50,000/QALY-gained threshold to establish a probability of being cost-effective, the results from the two scenario analyses do provide a good indicator for where the true cost-effectiveness of newborn screening for MLD vs. no screening is likely to lie. This is discussed more in the limitations and potential improvements to the decision analytic model outlined below. 

## 4. Discussion

This is the first study to evaluate the cost-effectiveness of adding newborn screening for MLD to the routine dried blood spot screening programme in the UK. The decision analytic model estimated that using an incidence rate of 1/100,000 live births, screening for MLD in the UK would identify 7 newborns with MLD per year, generating an incremental total gain of 246 discounted QALYs over the newborn cohort’s lifetime compared to no screening, whereas in current clinical practice only 2.6 children are identified in time to be eligible for treatment annually. Screening for MLD will increase the total discounted lifetime costs of the newborn cohort by an estimated £8.16 million; this is almost entirely driven by the cost of treating the additional four patients identified through NBS with arsa-cel (therapy and administration costs), who in the no-screening arm would not have been identified in time to be eligible for treatment and would have experienced the symptoms of MLD before dying prematurely. Using a discount rate of 1.5%, the base-case ICER for MLD screening vs. no screening is £33,212 per QALY gained, demonstrating that adding newborn screening for MLD to the current UK routine dried blood spot screening programme in the UK is a cost-effective use of NHS resources at the willingness-to-pay threshold appropriate for the severity and impact of MLD. The main drivers for these results are two-fold: (1) Firstly, screening allows the identification and treatment of all MLD babies, the majority of which in the no-screening arm are diagnosed too late for treatment—there are therefore substantial QALY gains for these patients in terms of both survival and quality of life. (2) Secondly, for some of the patients who were treated in the no-screening arm, some were either treated close to predicted onset of symptoms or already had early symptoms. So, whilst there are comparable survival gains, the quality of life in patients who have stabilised in GMFC-MLD health states with a lower perceived quality of life leads to lower QALY gains in the no-screening arm. Conversely, in the screening arm, all early-onset MLD patients are assumed to be treated within six to eight months of life. This means that all early-onset patients would be able to receive gene therapy potentially well before any sulfatides are able to accumulate and cause any irreversible damage leading to a reduced quality of life that we see in some patients in the no-screening arm.

The structure, assumptions and limitations in the Markov model that generated the long-term cost-effectiveness data for arsa-cel are similar to the economic models reported in Fahim et al. [26] and published health technology assessments from countries that have evaluated the cost-effectiveness of arsa-cel and in which it is reimbursed [11,27,28]. This suggests that the findings from this study are likely to be generalizable to other jurisdictions, particularly in countries where the willingness-to-pay thresholds are higher than in the UK, e.g., $150,000/QALY gained in the US. 

In the studies by Bessey and Weidlich investigating the cost-effectiveness of adding other rare neurodegenerative diseases to the DBS screening programme in the UK using similar economic model frameworks [14,19], the authors found that introducing screening for these conditions improved health outcomes and was less costly than no screening (i.e., screening dominated no screening). Whilst screening for MLD significantly improves health outcomes, it is not less costly compared with no screening. The difference between the findings of our study and those of other published studies looking at the cost-effectiveness of screening for a genetic condition is primarily due to the proportion of patients in each decision arm that receive treatment. In the studies by Bessey and Weidlich, all patients in both decision arms received treatment (and therefore incurred the cost); it was the timing of treatment in relation to pre- or post-disease onset that differed between the decision arms. Therefore, differences in costs and QALYs were attributable to the achievable health gains and cost savings due to lack of disease progression in the screening arm and not due to treatment costs. For MLD, screening results in significant QALY gains over no screening; however, due to the strict eligibility criteria required for treatment with arsa-cel, very few patients in the no-screening arm actually received treatment (and therefore incurred any treatment costs). Therefore, whilst implementing screening for MLD is cost-effective, it is not cost-saving. 

Recently published real-world evidence for the number of annual cases of MLD referred for potential treatment in the UK [13] showed that the number of referrals for gene therapy in a year exceeded the reported incidence rate of 1/40,000 [1], and far exceeds the average incidence that has been assumed for the base case in this study (1/100,000). It is likely that the higher number of patients in that study reflects the prevalent pool of MLD rather than the incident pool, given that many patients were symptomatic on referral and were of varying ages ranging from 17 months to 9 years. Increasing the number of MLD patients going through each arm of the decision model to these numbers results in incremental QALY gains similar to those reported in Horgan et al. Given that a higher incidence rate of MLD improves the ICER for screening vs. no screening, it can be inferred that the results from this study are a conservative estimate of the cost-effectiveness of newborn screening for MLD. 

There are some limitations associated with this decision analytic model. Firstly, for the late juvenile and adult-onset (LJ/AO) MLD input parameters, there are no published data for the long-term costs and benefits of HSCT treatment for MLD that could be used to inform the model; consequently, these parameters were informed by the HSCT cost and outcomes for an analogous disease. Real-life outcomes for these patients could be better or worse than assumed in the model; however, given that the percentage of LJ/AO incident patients is very small, even large changes in the costs and QALYs estimated for this group have minimal impact on the overall ICER for screening vs. no screening. In addition, because the data have been derived from the NICE appraisal of Strimvelis for ADA-SCID [23], the long-term costs of HSCT have been applied assuming that LJ/AO MLD patients will undergo HSCT as infants, the costs of which may differ from those for a teen or adult population. This is contrary to European consensus-based recommendations which suggest a deferred approach to treatment until subclinical evidence of disease onset is present. Finally, for the screening arm for the LJ/AO QALY inputs, an arbitrary disutility of −1.0 QALY has been applied to reflect the potential disutility of knowing from birth about having MLD that will develop in later life, as there was no published evidence to estimate this potential disutility. Removing this disutility improves the ICER for the LJ/AO subtype by £6242/QALY gained but only improves the overall ICER by £105, demonstrating that this potential disutility has minimal impact on the overall cost-effectiveness of screening. Furthermore, it could be argued that knowing about MLD earlier means that these late-onset patients could have earlier access to upcoming innovative treatments that they may not have if they were only diagnosed after having symptoms. More importantly, it gives patients and their families the option for treatment, which for the majority of MLD patients in current clinical practice is not possible. 

Whilst the impact of key input parameters in the decision tree have been tested in scenario analyses, particularly the probabilities at the chance nodes that were based on expert opinion, an improvement to the model structure would be to embed the Markov model in the decision tree rather than just use the results from the model as input parameters, so that probabilistic sensitivity analyses (PSA) could be performed for the long-term costs and outcomes. This would allow each input parameter to be varied within its specific distribution simultaneously to ascertain what proportion of the 10,000 iterations results in an ICER that falls below the £50,000/QALY-gained threshold, therefore establishing the probability of being cost-effective whilst capturing uncertainty in the input parameters. However, the results from this study provide a reliable indication of the cost-effectiveness profile for a screening programme for MLD and a probabilistic analysis, and more information is unlikely to yield findings that would contravene what has been observed here.

Another area for enhancement would be to broaden the perspective to a societal scope. The current study has been performed for the healthcare perspective, of most interest and direct relevance to the NHS. There are considerable additional potential benefits associated with treatment which have been captured in other accounts, e.g., productivity gains [7,26,29,30]. These will serve to improve the ICERs illustrated here. 

In conclusion, from a clinical point of view it is unequivocal that NBS for MLD has the potential to reduce the current high proportion of ineligible patients by ensuring rapid diagnosis at birth and timely treatment before symptom onset, while without NBS, the unfortunate and stark reality is that the majority of patients will be ineligible for treatment and will succumb to their disease due to a late diagnosis. Treating patients at an early age provides life chances, e.g., education and work-productivity benefits which are important considerations within the context of health technology appraisals. Whilst the feasibility of the MLD screening algorithm and the clinical rationale for screening for MLD have been evidenced in a number of publications, this study is the first to demonstrate that newborn screening for MLD is also a cost-effective use of NHS resources based on the outlined assumptions. These findings strongly support the inclusion of MLD in the official NBS program in the UK. 

## Figures and Tables

**Figure 1 IJNS-10-00045-f001:**
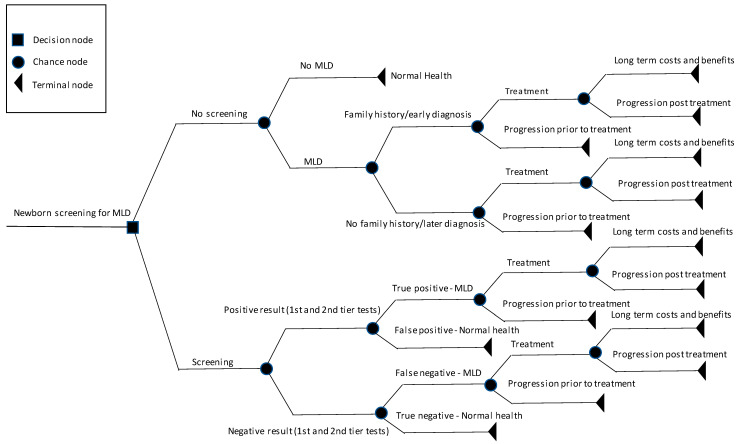
Schematic presentation of the decision analytic model depicting the screening and no screening arms for MLD.

**Figure 2 IJNS-10-00045-f002:**
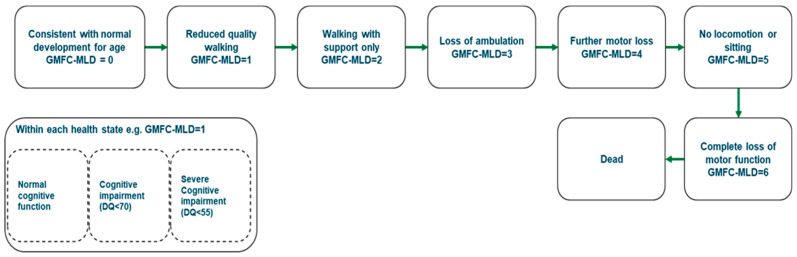
Model schematic of the eight-state partitioned survival and Markov framework model used to model long-term outcomes for patients with MLD in the NBS decision-tree model.

**Figure 3 IJNS-10-00045-f003:**
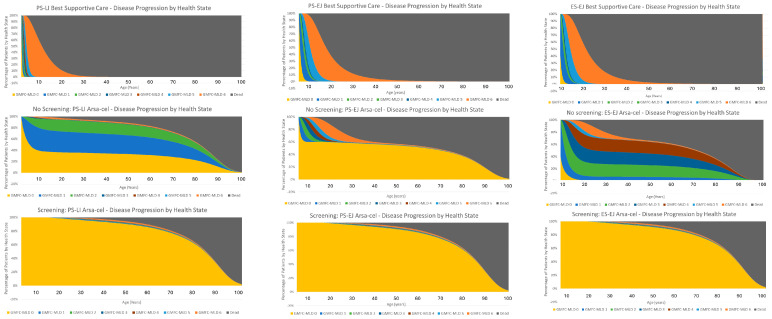
Graphical representation of the time spent in each GMFC-MLD health state for best supportive care and arsa-cel for late infantile MLD (**left panel**), pre-symptomatic early juvenile MLD (**middle panel**) and early-symptomatic early juvenile MLD (**right panel**) that were used to estimate the long-term costs and outcomes.

**Table 1 IJNS-10-00045-t001:** Epidemiology of MLD in the UK.

Epidemiology	Value	Source
Average annual number of births in England, Scotland and Wales 2016–2020	704,328	ONS (2021) [20]
Birth prevalence/incidence	1/40,000–1/160,000Mid-range of 1/100,000 used	*Orphanet*; Great Ormond Street Hospital for Children [1]; NICE HST18 scope [2]
Distribution of MLD Phenotypes in the UK:		
Late Infantile (LI)	60%	Clinical experts; Wang [8]
Early Juvenile (EJ)	30%	Clinical experts; Wang [8]
Pre-symptomatic early juvenile	15%	Clinical experts
Early-symptomatic early juvenile	15%	Clinical experts
Late juvenile/adult onset (LJ/AO)	10%	Clinical experts; Wang [8]

**Table 2 IJNS-10-00045-t002:** Probabilities for the chance nodes in the no screening arm of the decision tree model.

MLD Variant	Probability	Source
Probability of an MLD patient having no previous family history (NFH), e.g., index case:		
Late infantile	0.68	Clinical experts
Early juvenile	0.68	Clinical experts
Late juvenile/adult onset	0.6	Clinical experts
Probability of an MLD patient with NFH being diagnosed within treatment window for arsa-cel as per regulatory indication and for HSCT for LJ/AO:		
Late infantile	0.01	Clinical experts
Early juvenile (PS-EJ–ES-EJ)	0.17-0.19	Clinical experts
Late juvenile/adult onset	0.7	Clinical experts
Probability of an MLD patient having a family history (FH) of MLD, e.g., sibling of index case		
Late Infantile	0.32	Clinical experts
Early juvenile	0.32	Clinical experts
Late juvenile/adult onset	0.4	Clinical experts
Probability of MLD patient with a family history of MLD being diagnosed within treatment window for arsa-cel as per indication and for HSCT for LJ/AO:		
Late infantile	0.87	Clinical experts
Early juvenile	0.87	Clinical experts
Late juvenile/adult onset	0.9	Clinical experts

Abbrev: PS-EJ—Pre-symptomatic early juvenile; ES-EJ—early-symptomatic early juvenile; LJ—late juvenile; AO—adult-onset; NFH—no previous family history; FH—family history.

**Table 3 IJNS-10-00045-t003:** Probabilities for the chance nodes in the screening arm of the decision-tree model.

Input (Screen)	Probability	Source
Positive 1st-tier test (sulfatides)	0.0071	Hong et al. [15]
Positive 2nd-tier test (ARSA)	0.0005	Hong et al. [15]
False positive screen result	0.0000363	Hong et al. [15]
True positive screen result	0.9999637	Hong et al. [15]
False negative screen result	0.000001	Assumption as not quantifiable
True negative screen	0.999999	Assumption as not quantifiable
**Input (Outcome of screen)**		
Probability of disease progression prior to treatment	0.001	Clinical experts: As patients are identified at screening the likelihood that patients will experience rapid disease progression before treatment is extremely low.
Positive screen result probability of patient receiving treatment	0.999	
Probability of disease progression despite treatment	0.05	Clinical experts: As patients will be treated well before predicted onset of symptoms, the likelihood of disease progression once treated is very low.
Probability of achieving long-term outcomes from treatment	0.95	

The target population for the model was the average annual number of live births in the UK from 2016 to 2020.

**Table 4 IJNS-10-00045-t004:** Utility values used to generate QALYs for the long-term health outcomes in both arms of the decision model by GMFC-MLD health state for late infantile and early juvenile health states.

GMFC-MLD Health State	Late Infantile	Early Juvenile		
		Normal Cognitive Function (DQ > 70)	Moderate Cognitive Impairment (DQ < 70 and ≥55)	Severe Cognitive Impairment (DQ < 55)
GMFC-MLD 0	UK age-adjusted general population	UK age-adjusted general population	0.75	0.51
GMFC-MLD 1	0.71	0.90	0.65	0.42
GMFC-MLD 2	0.44	0.82	0.57	0.34
GMFC-MLD 3	−0.07	0.43	0.18	−0.05
GMFC-MLD 4	−0.22	0.12	−0.13	−0.36
GMFC-MLD 5	−0.35	0.05	−0.20	−0.43
GMFC-MLD 6	−0.47	0.01	−0.26	−0.49

**Table 5 IJNS-10-00045-t005:** Caregiver disutility by GMFC-MLD health state.

GMFC-MLD Health State	Number of Caregivers Required	Total Caregiver Disutility
GMFC-MLD 0	0	0
GMFC-MLD 1	0	0
GMFC-MLD 2	0	0
GMFC-MLD 3	1	−0.108
GMFC-MLD 4	1	−0.108
GMFC-MLD 5	2	−0.216
GMFC-MLD 6	2	−0.216

**Table 6 IJNS-10-00045-t006:** Monthly medical cost of care of an MLD patient by GMFC-MLD health state for children aged 0–18 and adults aged 19+.

Ages 0–18							
Cost Category	GMFC-MLD 0	GMFC-MLD 1	GMFC-MLD 2	GMFC-MLD 3	GMFC-MLD 4	GMFC-MLD 5	GMFC-MLD 6 *
Drugs	£0	£198	£229	£235	£235	£254	£263
Medical tests	£0	£156	£74	£74	£74	£76	£74
Medical visits	£0	£311	£307	£634	£680	£547	£558
Hospitalisations	£49	£78	£233	£350	£551	£641	£3658
GP and Emergency	£0	£9	£13	£15	£20	£23	£27
Healthcare equipment	£0	£34	£40	£76	£76	£91	£91
Respite Care	£0	£0	£0	£0	£0	£0	£0
Social services	£0	£0	£0	£0	£0	£0	£3864
**Total**	£49	£785	£897	£1385	£1636	£1632	£8535
**Ages 19+**							
**Cost Category**	**GMFC-MLD 0**	**GMFC-MLD 1**	**GMFC-MLD 2**	**GMFC-MLD 3**	**GMFC-MLD 4**	**GMFC-MLD 5**	**GMFC-MLD 6 ***
Drugs	£0	£198	£229	£235	£235	£254	£263
Medical tests	£0	£156	£74	£74	£74	£76	£74
Medical visits	£0	£311	£307	£634	£680	£547	£558
Hospitalisations	£0	£0	£156	£262	£454	£535	£3547
GP and Emergency	£0	£9	£13	£15	£20	£23	£27
Healthcare equipment	£0	£34	£40	£76	£76	£91	£91
Respite Care	£0	£0	£0	£0	£0	£0	£0
Social services	£0	£102	£481	£1311	£2417	£3523	£4679
**Total**	£0	£810	£1301	£2608	£3956	£5048	£9239

* The cost of care in GMFC-MLD 6 is based on a proportion of patients being cared for at home and a proportion being cared for in hospital.

**Table 7 IJNS-10-00045-t007:** Treatment and administration costs.

Early-Onset MLD		
List price for arsa-cel		£2,875,000 *
**Administration costs:**		
Item	Source	Value
Leukapheresis (cell harvest)	Weighted average of HRGs for stem cell (SA34Z) and bone marrow harvest (SA18Z). National Reference costs.	£4272
Conditioning	Hospitalisation for conditioning (4–7 days) based on clinical opinion and SmPC. HRG for paediatric metabolic disorder hospitalisation for non-elective inpatients (weighted average cost = £7761). Busulfan costs = £138 per patient; Busulfan 60 mg vial–8 pack = £367.81), average dose of Busulfan in clinical trials = 176.102 mg.	£7899
Administration and hospitalisation	HRG paediatric metabolic disorders admissions weighted average elective inpatient (weighted average cost = £5068). However, the SMPC states patient would stay about 4–12 weeks (average of 7.5 weeks) in the hospital, which is about 6 weeks longer than that reported for metabolic disorders inpatient admissions in Hospital Episode Statistics, which is 11 days (E75.2). The weighted average cost of elective inpatient excess bed day HRGs was calculated to be £460.73 (i.e., £5068/11). Overall hospital stay thus calculated as £24,188 (i.e., £5068 + [41.5 × 460.73]).	£24,188
Follow-up transplant costs **	Hettle et al [24].—NICE regenerative medicines report. 2017. Follow-up costs for allogeneic stem cell transplants. Discharge to 6 months = £28,390, 6–12 months = £19,502, 12–24 months = £14,073. Expert opinion is that follow-up for autologous transplant costs will be the same for allogeneic stem cell transplants, and patients will be discharged to metabolic care after two years.	£61,965
**Total administration costs:**		**£98,324**
**Late-onset MLD (lifetime costs, not just treatment cost and administration)**		
HSCT matched unrelated donor (MUD) total lifetime costs for ADA-SCID		£558,718
HSCT haploidentical total lifetime costs for ADA-SCID		£888,757
**Total lifetime costs assumed at terminal nodes of LJ/AO cohort:**		**£723,737**

* The negotiated price the NHS pays for arsa-cel was used in the economic analysis. ** Follow-up transplant costs were distributed evenly over the first two years of the model.

**Table 8 IJNS-10-00045-t008:** Number of MLD patients tracking through the decision-tree model framework.

		No Screening	Screening
MLD Phenotype	Number of Babies with MLD	Number of Babies Treated with Arsa-Cel (LI/EJ) or HSCT (LJ/AO)	Number of Babies Treated with Arsa-Cel (LI/EJ) or HSCT (LJ/AO)
LI—no family history	2.87	0.03	2.87
LI—family history	1.35	1.20	1.35
PS-EJ—no family history	0.72	0.12	0.72
PS-EJ—family history	0.34	0.29	0.34
ES-EJ—no family history	0.72	0.12	0.72
ES-EJ—family history	0.34	0.29	0.34
LJ/AO—no family history	0.42	0.30	0.42
LJ/AO—family history	0.28	0.25	0.28
All MLD cases	7.04	2.60	7.04

Abbreviations: LI—late infantile; PS-EJ—pre-symptomatic early juvenile; ES-EJ—early-symptomatic early juvenile; LJ/AO—late juvenile/adult onset; HSCT—haematopoietic stem cell transplant.

**Table 9 IJNS-10-00045-t009:** Total and incremental discounted costs and QALYs in the decision-tree model based on 704,328 screened babies.

	Total QALYs			Total Costs			ICER for Screening vs. No Screening (£/QALY Gained)
MLD Phenotype	No Screening	Screening	Difference	No Screening	Screening	Difference	
LI MLD only	1.39	162.69	161.31	£5,698,647	£11,084,772	£5,386,125	£33,391
PS-EJ MLD only	1.86	40.47	38.61	£1,674,332	£2,810,598	£1,136,266	£29,426
ES-EJ MLD only	−1.88	40.31	42.19	£1,301,133	£2,811,029	£1,509,896	£35,790
LJ/AO MLD only	13.43	17.05	3.62	£406,125	£533,447	£127,321	£35,173
General population health only	30,285,818	30,285,818	−0.07	0	£122,518	£122,518	-
All newborn babies	30,285,833	30,286,079	246	£9,080,237	£17,239,852	£8,159,615	£33,212

Abbreviations: LI—late infantile; PS-EJ—pre-symptomatic early juvenile; ES-EJ—early-symptomatic early juvenile; LJ/AO—late juvenile/adult onset.

**Table 10 IJNS-10-00045-t010:** Results from sensitivity analyses to test impact of varying key parameters on the cost-effectiveness results.

	All Newborn Babies			
Scenario	QALYs	Costs (£)	Scenario ICER	Difference from Base Case ICER
**Incidence Rate: 1/40,000**				
Screening	30,286,017	£41,922,816	£31,231	−£1981
No screening	30,285,401	£22,700,593	-	-
**Incidence rate: 1/160,000**				
Screening No screening	30,286,094	£11,069,104	£35,203	+£1990
	30,285,941	£5,675,148	-	-
**Discount rate: 3.5%**				
Screening	18,030,785	£17,144,509	£60,597	+£27,385
No screening	18,030,640	£8,374,776	-	-
**Distribution of MLD phenotypes: 45% LI, 25% PS-EJ, 25% ES-EJ, 5% LJ/AO**				
Screening	30,286,083	£18,624,338	£33,190	−£22
No screening	30,285,826	£9,436,156	-	-
**Family history of MLD probabilities—increase by 20%**				
Screening	30,286,083	£18,624,338	£34,891	−£1735
No screening	30,285,831	£10,153,754	-	-
**Family history of MLD probabilities—decrease by 20%**				
Screening	30,286,079	£17,902,240	£34,891	+£1679
No screening	30,285,828	£8,329,507	-	-
**Probability of disease progression prior to treatment increase from 0.001 to 0.1—screening arm only**				
Screening	30,286,061	£16,462,874	£32,148	−£1063
No screening	30,285,833	£9,080,237	-	-
**Probability of disease progression despite treatment increase from 0.05 to 0.2—screening arm only**				
Screening	30,286,041	£18,070,830	£43,345	+£10,133
No screening	30,285,833	£9,080,237	-	-
**Upper bound ICER: 20% reduction in incremental QALYs with 20% increase in incremental costs for MLD patients—screening arm only**				
Screening	-	-	£49,818	+£16,606
No screening	-	-	-	-
**Lower bound ICER: 20% increase in incremental QALYs with 20% decrease in incremental costs for MLD patients—screening arm only**				
Screening	-	-	£22,141	−£11,070
No screening	-	-	-	
**Screening costs only (a) +/−20% 1st tier and +/−80% 2nd-tier test; (b) 1st-tier cost increase from £0.14 to £0.71 per baby**				
a. Screening	30,286,079	£17,735,427	£32,995–£33,429	All ICERs within ±£217
No screening	30,285,833	£9,080,237		
b. Screening	30,286,079	£17,799,935	£35,492	+£2280
No screening	30,285,833	£9,080,237	-	-

## Data Availability

The original contributions presented in the study are included in the article/Supplementary Material, further inquiries can be directed to the corresponding author.

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
