# Peer review of "Exploring the Cost-Effectiveness of Newborn Screening for Metachromatic Leukodystrophy (MLD) in the UK"

_2409-515X, 2024, doi:10.3390/ijns10030045_

Round 1

Reviewer 1 Report

Comments and Suggestions for Authors

This paper reports on a cost-effectiveness analysis of newborn screening for metachromatic leukosdystrophy, and shows favourable cost-effectiveness. The analysis is based on decision trees with screening and no screening arms.  These decision trees are described adequate. However, the results of the analysis are heavily depending on the model inputs on the long term outcomes. For these input parameters the information is described but no values are given. This makes it impossible to assess the scientific soundness of the analysis. 

Other comments:

- In figure 1 I would expect a possibility of family history/early diagnosis for infants with a false negative result?

- table 3: is 'the positive screen result probability of patient receiving treatment' meant to be the complementary probability of disease progressions prior to treatment. These seem to me different things. 

- what is the exact role of the Markov model in this analysis? Is the Markov model embedded in the decision tree or were results from this Markov analysis only used as input to the decision tree model?

- page 8 line 268: which 24 health states are meant here? 7 gross motor function stages combined with 3 cognitive ability stages results in 21 states? Can the utility values from reference 19, which seems to be only a conference abstract, be included in the article, e.g. as an appendix?

- page 9 line 334-344, please include values for the costs mention in the paragraph in the article, e.g. as an appendix. 

- table 5: Where does the cost in a screening situation of GBP 417,034 come from, if the cost of screening itself is only GBP 122,518?

- table 5: also include sensitivity analyses on the probabilities of disease progression prior to treatment and despite treatment

Reviewer 2 Report

Comments and Suggestions for Authors

Brief summary

The aim of this paper is to assess the cost-effectiveness of newborn screening for metachromatic leukodystrophy (MLD) in the UK. This appears to be the first paper addressing the cost-effectiveness of newborn screening for MLD.

General concept comments

The model structure of a 8-state partitioned survival framework and Markov structure following the presented decision tree is appropriate for newborn screening and the condition

Details are given of the screening parameters and they appear to be justified and appropriate given the early pilot data. However, it is not clear if all false positive cases are excluded based solely on the DBS or if they need to attend for confirmatory testing and would therefore incur a cost of an outpatient appointment, confirmatory testing, and an appointment to discuss the results. Given that a decrement of 0.01 is applied to a false positive case this would imply that they are informed of the result and therefore should incur some further costs. Although this will have a minor impact on the results.

I have 3 main concerns with the manuscript that need to be addressed.

Firstly, the majority of the parameters used in the model were not provided which makes it impossible to comment on the appropriateness of the model methodology. It also makes it difficult to understand the model and what is driving the results. The manuscript should report enough details that the results could be reproduced based on the manuscript.

A full parameters table needs to be included as part of the manuscript. This should include the health state QALYs and costs, treatment costs, transition probabilities for the three MLD subtypes and by responder type.

It would also be helpful to have a table or matrix clearing defining the different subgroups and how they relate to the responder types and treatments. I wasn’t clear how the Early Juvenile group is split into the pre-symptomatic group and early symptomatic group? I would assume that the presymptomatic group would all be from the family history group but this is not the case in Table 4? Could this be clarified?

Could you clarify why in table 4 there are there lower numbers of MLD patients treated in the no screening family history arm than in the screen family history arm? Surely this should be the same if they are both identified pre-symptomatically.

In the introduction it states that there are four clinical forms based on the age of presentation. Following screening treatment will be started prior to the development of symptoms. How are the different clinical types identified in the screening arm/family history group? Could this be clarified in the text.

It is unclear at what age the Late juvenile/adult onset group receive treatment? If they are screened will they receive HSCT as an infant/child or will they wait until they are older. The costs used in the model are based on the HST for Strimvelis in which infants undergo HSCT and the costs may differ for an adult population.

Secondly, the manuscript in many places refers to NICE guidelines in terms of the discount rate and cost-effectiveness threshold. In the UK NICE does not make decision regarding newborn screening programmes. The UK National Screening Committee (UK NSC) makes the decision on whether a new screening programme should be recommended and will consider the cost-effectiveness evidence. Therefore, the discount rate and thresholds generally considered by the UK NSC should be included in the manuscript. A discussion of how they differ and the implications – such as the higher threshold used in the NICE HST appraisals could be included in the discussion.

While there is an argument the treatment meets the criteria for a lower discount rate, especially if presymptomatic treatment increases the effectiveness of the treatment.  I note that in the NICE guidance for Atidarsagene autotemcel “…the committee considered that the non-reference discount rate of 1.5% was not appropriate for decision making.” Therefore in this manuscript the standard 3.5% discount rate should have been used in the basecase with a sensitivity analysis using 1.5%.

Thirdly, no probabilistic sensitivity analysis has been undertaken. It is standard in cost-effectiveness analysis that a PSA is undertaken and reported.  Given the inherent uncertainties in this case – the condition is rare, there is limited long term outcome data, - understanding the uncertainty around the results is crucial. It would also be helpful for other results to be presented such as lifeyears, time spent in each health state, and/or total costs broken down into screening costs, treatment costs, health state costs.  This would make it easier to understand what the benefit of screening is and which assumptions are driving the results

Specific comments

It would be helpful to have abbreviations below each table

Table 4  -  For the ES-EJ phenotype both rows refer to no family history

Reviewer 3 Report

Comments and Suggestions for Authors

The discussion of the natural history and treatments for MLD is well written and informative. In contrast, the documentation of the assumptions underlying the cost-effectiveness model is incomplete and lacks transparency.

L266-276: The description of methods for the calculation of QALY losses with MLD is inadequate. The only reference cited is a conference abstract from 2020. Given the importance of QALY estimates for the assessment of cost-effectiveness, it is essential that the utility elicitation study be subject to objective peer review. It is unclear why the sponsors have not done so in the past 4 years. At a minimum, for the cost-effectiveness analysis to be peer reviewed it is essential that the manuscript documented the utility elicitation study be included as a supplement. Furthermore, the authors fail to provide appropriate context and support for the magnitude of the disutility associated with advanced MLD. The only reference that is cited, number 8, appears irrelevant.

L318-333: The authors assume a GBP 0.174 per infant marginal cost of adding MLD to UK screening panel is, which seems rather low and may be overly optimistic. The bulk of this estimate, GBP 0.14 per baby, was derived from a previous CEA of expanding NBS to include five IEMs, taking the estimated laboratory staffing cost and dividing by five. It implicitly assumes zero cost for equipment and supplies. The authors should provide corroboration from the North German screening program that their assumptions are justified.  

L334-343: The text describes in broad terms the methods used to project costs of care for patients with MLD but does not report actual estimates. At a minimum, the authors should report estimates of annual HCRU costs for each of the GMFC-MLD states.  

Comments on the Quality of English Language

Copy editing comments

L138: delete “in” at end of line

L141: replace “are” with “is” because the subject, “importance” is singular.

L174: Specific, not specifically

Round 2

Reviewer 2 Report

Comments and Suggestions for Authors

The authors have addressed my key concerns with the manuscript. They have included substantially more detail on the model parameters and model assumptions. While no PSA has been included the additional sensitivity analyses do provide more detail on the key uncertainties within the model and there is increased discussion of the model limitations.   

I have three minor comments

1.       Could the discount rate sensitivity analysis be included in Table 9. It is referred to in the text but it would be clearer if it was also included in the table.

2.       On figure 3 could the age on the X axis be changed from months to years to improve clarity

3.       It would be worth highlighting in the discussion that it is only cost-effective if the severity modifiers are used to increase the threshold to £50,000 and if a discount rate of 1.5% is used. The acceptance of both of these would, I believe, be at the discretion of the committee based on the evidence presented - I’m not sure if severity modifiers have been used in a decision by the NSC yet.

Author Response

Comment 1: Could the discount rate sensitivity analysis be included in Table 9. It is referred to in the text but it would be clearer if it was also included in the table.

Of course, this sensitivity analysis has been included in Table 9.

Comment 2: On figure 3 could the age on the X axis be changed from months to years to improve clarity.

Apologies the X axis wasn’t easy to read! The model has a monthly cycle to capture the rapidly progressive phase of the disease from GMFC-MLD 2 to 5 and so the proportion of patients in each GMFC-MLD state throughout the lifetime horizon of the model were generated monthly. Figure 3 is a direct output from the Markov engines, which is why age was recorded in months. The Figure has now been amended to age in increments of 10 years to make it easier to read.

Comment 3: It would be worth highlighting in the discussion that it is only cost-effective if the severity modifiers are used to increase the threshold to £50,000 and if a discount rate of 1.5% is used. The acceptance of both would, I believe, be at the discretion of the committee based on the evidence presented - I’m not sure if severity modifiers have been used in a decision by the NSC yet.

Thank you for the comment and you are right, I don’t think severity modifiers have been used explicitly in a decision by the NSC yet. However, I believe there is a willingness on the NSC’s behalf to explore the potential, as in the June 2022 minutes examining the evidence base for screening for TYR1, there was a good deal of discussion about the use of NICE’s health technology evaluation modifiers to allow greater flexibility that enables NICE to justify a higher willingness to pay threshold. But we do recognise that the acceptance of both the 1.5% discount rate and an increased threshold up to £50,000/QALY would be at the discretion of the NSC – some wording to that affect has been added to the methodology section 2.4 WTP Thresholds.   

With regards to the first point, lines 574-578 in the Discussion section state both the discount rate and the WTP threshold when discussing whether screening for MLD is a cost-effective use of NHS resources. In addition, the base case ICER (£33,212/QALY gained) is reiterated, which is above the traditional £20,000-£30,000/QALY threshold. Some wording regarding the WTP threshold has also been added to the abstract to put the statement about screening for MLD being a cost-effective use of NHS resources into context (the 1.5% discount rate is already mentioned). 

Reviewer 3 Report

Comments and Suggestions for Authors

The authors were fully responsive to the comments. The authors have gone above and beyond the usual expectations of CEAs. For example, the inclusion of spillover effects of advanced MLD on parental health is an important contribution. It could be interesting (but not required) to report in a sensitivity analysis the impact of this variable on the ICER, similar to the analysis that compares 1.5% and 3.5% discount rates. 

The authors might wish to cite a May 1 JAMA viewpoint by David Rind that cited the ICER report that affirmed that the MLD gene therapy is highly effective and likely to be cost-effective in the United States. "Curative gene therapies for neuromuscular disorders affecting young children can be extremely valuable and worth very high prices. The Institute for Clinical and Economic Review (ICER) felt that Zolgensma, a gene therapy for spinal muscular atrophy, could be worth up to $2.1 million and that OTL-200, a gene therapy for metachromatic leukodystrophy, could be worth up to $3.9 million.7,8"

Comments on the Quality of English Language

The English is readily comprehensible, but syntax could be improved in some instances. I spotted one typo, with the s and h in thresholds inverted.  

Author Response

Comment 1: The authors were fully responsive to the comments. The authors have gone above and beyond the usual expectations of CEAs. For example, the inclusion of spillover effects of advanced MLD on parental health is an important contribution. It could be interesting (but not required) to report in a sensitivity analysis the impact of this variable on the ICER, similar to the analysis that compares 1.5% and 3.5% discount rates. 

Thank you very much for your comments and we are pleased that your first-round comments have been addressed.

With regards to your comment regarding what the impact would be on the ICER from the spillover effects of advanced MLD disease on parental health - removing this caregiver disutility worsens the ICER (circa £1,000/QALY gained) as it increases the QALY gains in the no screening arm. This is because the majority of MLD patients in the no screening arm do not receive treatment and so progress to the later stages of MLD incurring the worsening caregiver disutility, therefore reducing the total QALYs. Without this disutility, the total QALYs are higher and so the incremental gains for screening are smaller. The effect isn’t as pronounced as you might anticipate because the majority of patients have late infantile MLD and die young, so the disutility does not last very long in comparison to the model time horizon.

Along the same vein, an important consideration that could have been included in this analysis with regards to spillover effects is the negative impact on parental/familial HRQoL of losing a child and the number of years post death that the impact of this bereavement persists. In most models I have come across, the impact of the disease on the family ends at point of death. I know Tufts have done some research into this area for SMA and found a significant impact on HRQoL at 2 and 5 years after losing a child, but not at 10 years. I think these points could potentially be controversial so we have not included them in the manuscript but thought you might be interested in the concept.

Comment 2: The authors might wish to cite a May 1 JAMA viewpoint by David Rind that cited the ICER report that affirmed that the MLD gene therapy is highly effective and likely to be cost-effective in the United States. "Curative gene therapies for neuromuscular disorders affecting young children can be extremely valuable and worth very high prices. The Institute for Clinical and Economic Review (ICER) felt that Zolgensma, a gene therapy for spinal muscular atrophy, could be worth up to $2.1 million and that OTL-200, a gene therapy for metachromatic leukodystrophy, could be worth up to $3.9 million.7,8"

Many thanks for the recommendation to cite the JAMA viewpoint by David Rind which refers to the ICER report of arsa-cel. The academic editor has instead suggested that we refer to the published article by Fahim et al. in the Journal of Managed Care Spec Pharm. 2024 Feb; 30(2): 201–205 – which essentially summarizes the ICER analysis of arsa-cel but in more detail.

Consequently, we have added some wording to the discussion around the use of similar Markov models to evaluate the long-term cost-effectiveness of arsa-cel in other jurisdictions outside of the UK in terms of similar model structure, limitations and results - which hopefully takes this consideration on board and highlights the generalizability of our findings to other countries.